# Memristive Devices from CuO Nanoparticles

**DOI:** 10.3390/nano10091677

**Published:** 2020-08-26

**Authors:** Pundalik D. Walke, Abu ul Hassan Sarwar Rana, Shavkat U. Yuldashev, Verjesh Kumar Magotra, Dong Jin Lee, Shovkat Abdullaev, Tae Won Kang, Hee Chang Jeon

**Affiliations:** 1Nano Information Technology Academy, Dongguk University, Seoul 04620, Korea; giteedig@gmail.com (P.D.W.); shavkat@dongguk.edu (S.U.Y.); birju.srm@gmail.com (V.K.M.); jin514rin@naver.com (D.J.L.); abdshov77@gmail.com (S.A.); twkang@dongguk.edu (T.W.K.); 2Intelligent Mechatronics Engineering/Smart Device Engineering, Sejong University, Seoul 05006, Korea; rana@sejong.ac.kr; 3Department of Physics, National University of Uzbekistan, Tashkent 100174, Uzbekistan

**Keywords:** CuO nanomaterials, negative differential resistance, Poole–Frenkel conduction, switching ratio, resistive switching, space charge limited current

## Abstract

Memristive systems can provide a novel strategy to conquer the von Neumann bottleneck by evaluating information where data are located in situ. To meet the rising of artificial neural network (ANN) demand, the implementation of memristor arrays capable of performing matrix multiplication requires highly reproducible devices with low variability and high reliability. Hence, we present an Ag/CuO/SiO_2_/p-Si heterostructure device that exhibits both resistive switching (RS) and negative differential resistance (NDR). The memristor device was fabricated on p-Si and Indium Tin Oxide (ITO) substrates via cost-effective ultra-spray pyrolysis (USP) method. The quality of CuO nanoparticles was recognized by studying Raman spectroscopy. The topology information was obtained by scanning electron microscopy. The resistive switching and negative differential resistance were measured from current–voltage characteristics. The results were then compared with the Ag/CuO/ITO device to understand the role of native oxide. The interface barrier and traps associated with the defects in the native silicon oxide limited the current in the negative cycle. The barrier confined the filament rupture and reduced the reset variability. Reset was primarily influenced by the filament rupture and detrapping in the native oxide that facilitated smooth reset and NDR in the device. The resistive switching originated from traps in the localized states of amorphous CuO. The set process was mainly dominated by the trap-controlled space-charge-limited; this led to a transition into a Poole–Frenkel conduction. This research opens up new possibilities to improve the switching parameters and promote the application of RS along with NDR.

## 1. Introduction

A new era of non-volatile resistive memory devices has emerged due to the physical limitations of existing memory devices. Due to their scope, memristors have drawn substantial attention in the previous decades. Several specific properties make memristors a favorable and promising candidate for non-Boolean neuromorphic computing [1]. Some of its most common attributes include low power consumption, high scalability, multiple switching states and non-damaging readout. Long retention characteristics allow memristors to take over the complementary metal-oxide-semiconductor (CMOS) and to integrate it with CMOS technology [2,3]. Negative differential resistance (NDR) along with resistive switching can have additional applications such as resonant tunneling transistors [4], high-frequency oscillators, [5] memory devices, and multi-level logic devices [6]. The NDR effect in negative bias region reduces variability in reset, which results in a controlled reset.

Memristors are two-terminal memory cells with a sandwiched active layer. This simple two-terminal structure has shown a promising output in many metal oxides such as TiO_2_ [1], NiO [2], ZnO [7], perovskites [8], and transition metal di-chalcogenide monolayers [9,10]. Among all oxides, CuO is the most widely investigated and reported material because of its good availability, easy synthesis, good reliability, non-toxic behavior, and low cost. CuO has recently been explored for memristive devices with a superior retention time, low power consumption, good compatibility with CMOS technology [11], and excellent endurance [12,13,14,15,16]. This gives CuO substantial advantages over other materials.

The NDR effect found several applications in high-frequency oscillators and multi-level switching. Hence, the industry is focused on understanding, analyzing, and discovering the NDR effect along with resistive switching. Several reports have been published on NDR with ZnO [17], polymers, graphene oxide, nanocomposites [1,17,18,19,20,21], other transition metal oxides (TiO_x_ [22], and FeO_x_ [23]). However, the NDR effect is rarely reported in CuO. Recently Kadima et al. demonstrated an NDR device with ZnO nano-rod arrays [24].

Among all switching mechanisms, the conductive filament (CF) model is widely analyzed to explain switching. However, the CF model has some shortcomings. The reset in the CF model involves the rupture of filaments making a scattering voltage distribution. This deteriorates the device endurance and leads to variability in switching. 

In this study, we investigate the effect of native oxide on resistive switching, which reduced the variability and demonstrated the hysteresis current–voltage (I-V) characteristics along with NDR. We used an ultra-spray pyrolysis method to deposit amorphous CuO nanoparticles on *p*-Si to obtain a multifunctional device with resistive switching and NDR properties. The topology of the device was well defined by scanning electron microscopy (SEM). The phase purity of CuO nanoparticles was confirmed by Raman spectroscopy. The resistive switching and negative differential resistance were obtained by I-V characteristics. To understand the conduction mechanism, temperature-dependent I-V characteristics were carried to substantiate the space-charge-limited current conduction (SCLC). The log-log plot further verifies the SCLC. The prime goal of this study was to reduce the variability in switching and to reduce power consumption by introducing the native oxide layer. Here, we fabricated Ag/CuO/ITO to substantiate the effect of native oxide. I-V characteristics of Ag/CuO/ITO were compared with that of Ag/CuO/SiO_2_/p-Si. The heterostructure device improved the resistive switching (RS) by reducing the switching variability. This multifunctional device has great potential for advanced multifunctional non-volatile memories.

## 2. Materials and Methods 

The sample preparation used 0.1 M of an aqueous solution of copper acetate monohydrate (Cu(CH_3_COO)_2_.H_2_O) from Alfa Aesar (CAS: 6046-93-1) with 98% purity. The sample was made by continuously stirring with a magnetic stirrer for 15 min at 80 °C. The precursor solution was taken in the precursor reservoir containing the ultrasonic sensor. The carrier gas pressure was set via a pressure knob. The chamber temperature was set to 350 °C via a temperature controller [25]. The precursor solution was sprayed using a jet nozzle using air as the carrier gas on to the pre-heated p-type (100) Si substrates with a carrier concentration of 10^20^ cm^−3^. The substrate was not cleaned to protect the native oxide layer. Native oxide is usually 5 Å to 6.7 Å [26]; however, further growth of the oxide was evident during the deposition in an oxygen-rich environment. Ag was deposited as a top electrode using an e-beam metal evaporator (Korea vacuum tech., ltd, Gyeonggi-do, Korea). Figure 1 shows the schematics including a step-by-step procedure of device fabrication.

## 3. Characterization

This section describes the characterization techniques used to analyze the active CuO layer. The morphological evolution of deposited CuO nanoparticles was analyzed using scanning electron microscopy (SEM) (FESEM, Philips, Model: XL-30, Amsterdam, The Netherland). The elemental properties such as weight and atomic percentage of individual elements in a compound were characterized by energy-dispersive X-ray spectroscopy (EDS) connected with the FESEM equipment (FESEM, model: JSM-6701F, JEOL, Japan). Optical measurements of the CuO nanoparticles were performed using a Raman spectrometer (Princeton Instruments, Spectra Pro 2500i) at an excitation wavelength of 525 nm with a 0.5-s shutter speed. Electrical measurements were investigated using a Keithley 617 programmable electrometer parameter analyzer (Tektronix, Model: Keithley 617, Beaverton, OR, USA).

## 4. Results and Discussion 

Figure 2a shows the SEM image of CuO nanoparticles deposited by ultra-spray pyrolysis (USP). The CuO nanoparticles are uniformly deposited on the SiO_2_/p-Si substrate. The size of CuO nanoparticles is from 20 nm to 200 nm with an average size of 110 nm. Typical Raman peaks were observed at 268 cm^−1^, 305 cm^−1^, and 605 cm^−1^ as shown in Figure 2b [25]. These Raman peaks are assigned to the symmetry of A_g_(1), B_g_(1), and B_g_(2) modes of CuO, which confirms the CuO phase formation.

The I-V characteristics in Figure 3a clearly show the hysteresis curve, which demonstrates the non-volatile resistive switching along with negative differential resistance. The voltage was swept from 0 » 3 V » 0 » −3 V » 0. A stable resistive switching (RS) is observed when the bias voltage sweeps from 0 V to 3 V; the current switches from 10^−6^ A to 10^−3^ A at a set voltage of 1.7 V. The device maintains a low resistive state (LRS) when the bias voltage sweeps back from positive (3 V) to negative (−0.7 V), NDR is observed at −0.8 V, and the device switches off. Figure 3b demonstrates repeatability in switching up to 50 cycles, emphasizing the reproducibility and stability of the device. The inset in Figure 3b shows a clear cut of the NDR. The current decreases sharply with an increase in potential. Figure 3c shows the endurance performance of the device. Resistance was taken as a function of the number of cycles, which show the consistency and stability in low resistive state (LRS) and high resistive state (HRS). The semi-log I-V curves in Figure 3d give a more accurate picture of NDR and RS with a switching ratio of 10^3^.

Figure 4 presents the response of the device at different sweep rates. The device shows a stable response at higher sweep rates and performs well. The presence of trap charges in native oxide provides triggering of the next switching cycle at the same location, which improves the response.

To understand the mechanism, we implemented and proved the amorphous model. The traps generated by localized states in amorphous CuO nanoparticles film had a strong influence on the injected current in response to applied voltage [27]. The interaction of the injected carriers in defect states affects the magnitude of current, which also affects the current–voltage characteristics.

To prove the RS and NDR mechanism and investigate the role of traps, Log V vs. Log I and temperature-dependent I-V characteristics were plotted in Figure 5a,b, respectively. The current was proportional to voltage (*IαV*) at a lower voltage. Hence, the space-charge-limited current (SCLC) was close to negligible, and Ohm’s law dominated the I-V characteristics [2,28]. The transport is explained by the presence of thermal equilibrium free carriers given by Equation (1):*n* = *N* [exp [(*E* − *E_c_*/*kT* )]](1)

Here, *n* is a free carrier, *N* are the effective density of states in the conduction band, *E_C_* is the energy at the bottom of the conduction band, *k* is Boltzmann’s constant, and *T* is the temperature. As the voltage increases further, the SCL current becomes obvious as seen in the slope via *IαV* (Equations (2) and (3)) [28]. Hence, we inferred that the conduction was dominated by the shallow trap SCL current. The shallow trap square law is given by:
(2)$$J\;=\;\frac{9}{8}\;µ\varepsilon \frac{V^{2}}{L^{3}}$$
where *V*, *L*, μ and ε are the voltage, the distance between Top Electrode and Bottom Electrode, the free-charge mobility, and the dielectric constant, respectively. As the applied field further increases, the strong potential caused the effective depth of a coulombic-attractive trap to be reduced due to the Poole–Frenkel (PF) effect [14,28]. The reduction in depth reduces the barrier height resulting in an increase in the current level. This steep jump in current is higher than that predicted by the standard SCL current theory. The steep increase in the current is obvious in Figure 5a. The shallow trap SCL current density equation incorporating the PF effect can be given as:(3)$$J\;=\;\mu \Theta \varepsilon \xi \exp\left(\frac{\beta \surd \varepsilon }{kT}\right)\frac{d\varepsilon }{dx}$$
where *Θ* is the ratio of free to trapped charge concentration, ξ is an electric field and *β* is the Poole–Frenkel constant. Trap-induced space-charge-limited current was further proved by plotting temperature-dependent I-V in Figure 4. The data suggested that the change in temperature led to a variation in the trap distribution. The voltage dependence of space-charge-limited current is given by
(4)$$I\;\alpha \;V^{(T_{C}/T)\;+I}$$
where *T_c_* is a characteristic temperature describing the distribution of traps in energy. An increase in temperature does not alter the total amount of space charge but does increase the fraction of space charge in the conduction band. Equation (4) indicates that the temperature should be shifted in the current–voltage curve along with the voltage axis toward lower voltages [29]. Beyond room temperature, the switching and NDR effects were not observed over 25 °C.

The tunneling current had a negative temperature coefficient, which proved the tunneling current is dominated by the conduction process in the negative region. Based on the mechanism, the RS can be described by the formation of an Ag conductive filament due to a trap control charge transfer enabling the abrupt switching from HRS to LRS [24]. Ag is an active metal, and the Ag atoms are possibly ionized for the Ag/CuO/SiO_2_/p-Si device. The Ag atoms can be ionized into Ag ions with the electric field. The electric field drives the Ag ions into defects level of CuO nanoparticles. When the Ag ions accumulate to a certain extent, the conductivity of the material will increase substantially because the Ag ions play the role of conductive filaments to finish the set process leading to the formation of conductive paths [21].

To further investigate the role of native oxide, p-Si, and Ag, we fabricated a device consisting of ITO as the bottom electrode. Figure 6 shows the I-V characteristics of the device with Ag as a top electrode and ITO as a bottom electrode with CuO nanoparticles acting as an active layer. In the absence of SiO_2_, the device showed resistive switching without NDR. The switching was described by the formation of Ag filament based on the linear I-V characteristics. The device did not show NDR in the absence of oxide. The abrupt uncontrollable switching also characterizes the absence of native oxide. The results showed no NDR in the case of Ag/CuO/ITO, and we inferred that NDR in Ag/CuO/SiO_2_/p-Si was mainly due to electric field-induced charge transfer between CuO and SiO_2_; the high on/off ratio was due to the oxide barrier that contributes to an increase in the off-state resistance.

Considering all the above results, we propose a model to describe the NDR and RS behaviors shown in Figure 7a–e. The absence of NDR in Ag/CuO/ITO provided a piece of evidence that NDR was due to oxide-trapped charges associated with defects in SiO_2_.

The device is initially HRS due to traps in the CuO nanoparticles and the Schottky barrier at the Ag/CuO/SiO_2_/Si interface. Initially, the Schottky barrier at the SiO_2_/p-Si interface reduces when a forward voltage sweep is applied, resulting in an injection of charges. These injected charges are then trapped in oxide interfacial traps [30]. Simultaneously, the electric field drives the Ag ions into defect levels of CuO nanoparticles. This initiates the forward conduction as shown in Figure 7a. The filled interfacial traps reduce oxide resistance [31]. Further increases in the electric field accumulate Ag ions, which then play the role of conductive filaments. Therefore, the current increases sharply with further increases in the electric field due to CF; the filled traps are shown in Figure 7b. The device remains in an LRS until a negative voltage is applied to rupture the CF. The trapped electrons are then released under bias (Figure 7c). With a further increase of the negative voltage, the charge injection from the Ag electrode is prevented by the Schottky barrier at the interface. The current decreases gradually upon increasing the sweeping voltage. This explains the NDR behavior (Figure 7d). The current decreases until the trapped electrons are released completely (Figure 7e).

## 5. Conclusions

The Ag/CuO/SiO_2_/p-Si memristive device improves the switching characteristics by reducing the variability. It was successfully demonstrated by comparing the switching response without the native oxide. The traps generated by localized states in CuO nanoparticles control charge transfer and enable abrupt switching from HRS to LRS leading to a high switching ratio. The NDR phenomena were explained by the Schottky barrier and trapping/detrapping in interfacial traps in SiO_2_. The I-V characteristics on a device without oxide were investigated to determine the effect on NDR. The temperature-dependent I-V characteristics prove that the conducting mechanism consists of space-charge-limited conduction (SSLC). The combined mechanism includes SCLC, conductive filament (CF), and trapping/de-trapping in the native oxide layer: these features result in RS and NDR. In short, a heterostructure device with Ag/CuO/SiO_2_/p-Si gave highly stable non-volatile RS along with NDR that facilitates many applications.

## Figures and Tables

**Figure 1 nanomaterials-10-01677-f001:**
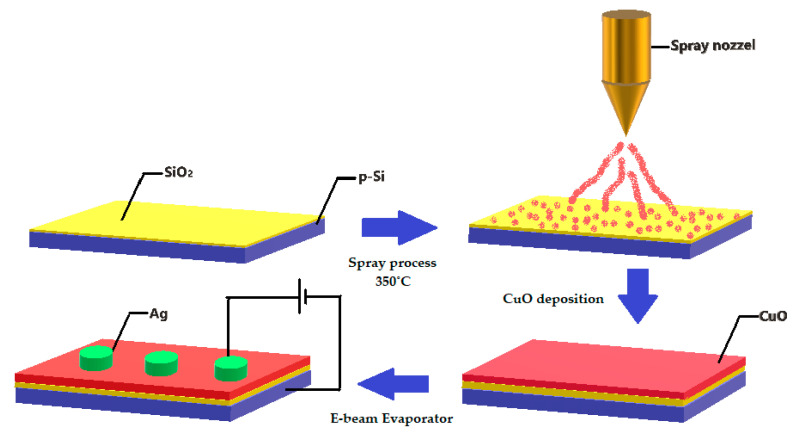
Step-by-step schematic representation of Ag/CuO/SiO_2_/p-Si device fabrication.

**Figure 2 nanomaterials-10-01677-f002:**
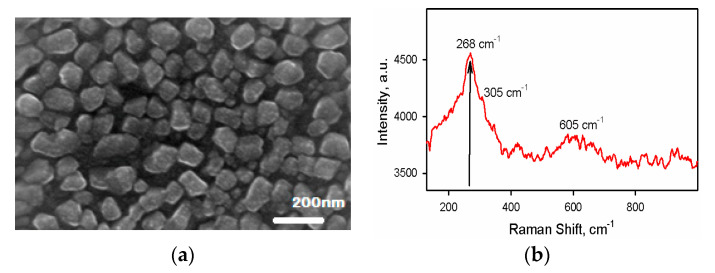
(**a**) SEM image of a device showing CuO nanoparticles uniformly distributed. (**b**) Raman spectra of deposited CuO nanoparticles showing peaks at 268 cm^−1^, 305 cm^−1^, and 605 cm^−1^.

**Figure 3 nanomaterials-10-01677-f003:**
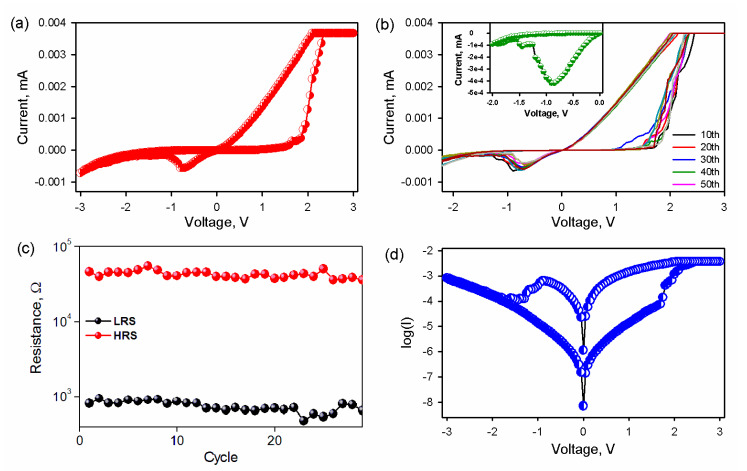
(**a**) Current–voltage (I-V) curve of Ag/CuO/SiO_2_/p-Si resistive switching and (**b**) multiple resistive switching up to 50 cycles depicting reproducibility. The inset shows the formation of negative differential resistance (NDR), (**c**) endurance performance of Ag/CuO/SiO_2_/p-Si, and (**d**) semi-log current–voltage characteristics of a multilayer Ag/CuO/SiO_2_/p-Si showing NDR and on-off conductance of 1 × 10^3^ in the positive region. The voltage was swept from −3 V to 3 V. The V_set_ and V_reset_ are 1.5 V and −0.65 V.

**Figure 4 nanomaterials-10-01677-f004:**
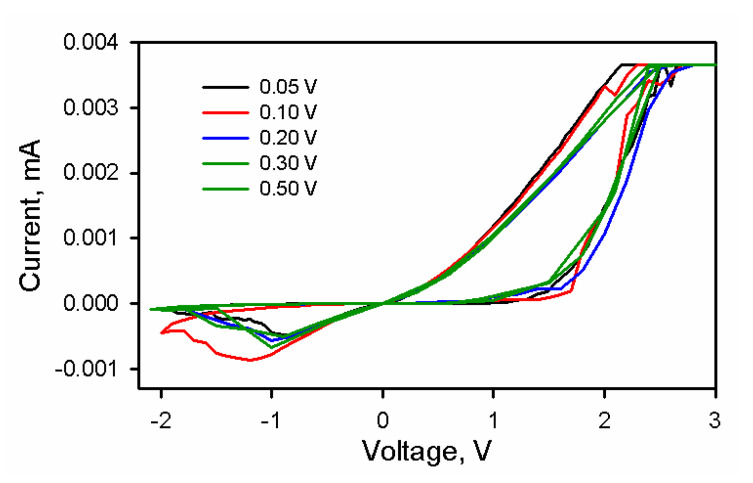
The I-V characteristics at multiple sweeping rates.

**Figure 5 nanomaterials-10-01677-f005:**
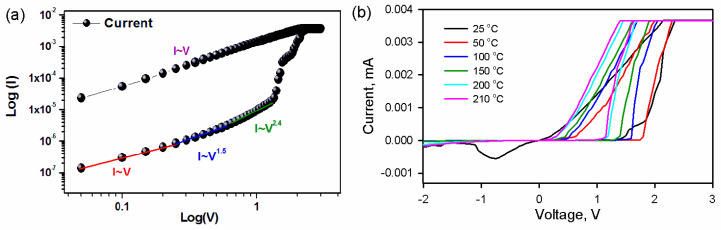
(**a**) The logarithmic I-V curve in the positive bias region and (**b**) temperature-dependent current–voltage characteristics of a multilayer Ag/CuO/SiO_2_/p-Si. Device temperature was varied from 25 °C to 210 °C.

**Figure 6 nanomaterials-10-01677-f006:**
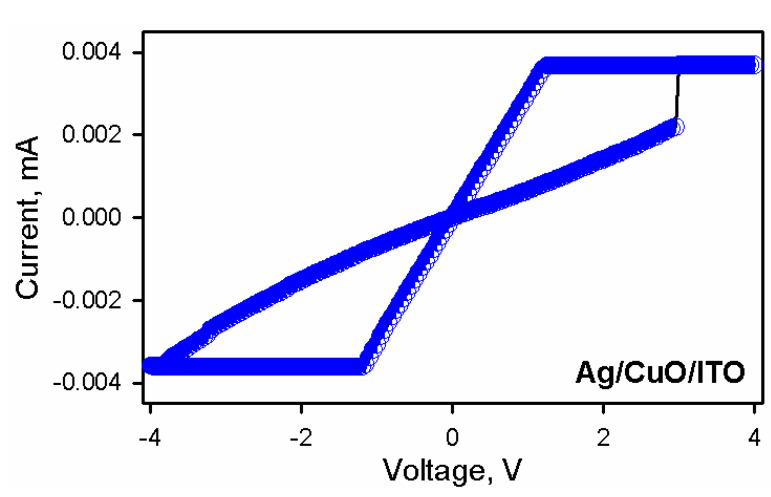
The I-V characteristics of Ag/CuO/ITO.

**Figure 7 nanomaterials-10-01677-f007:**
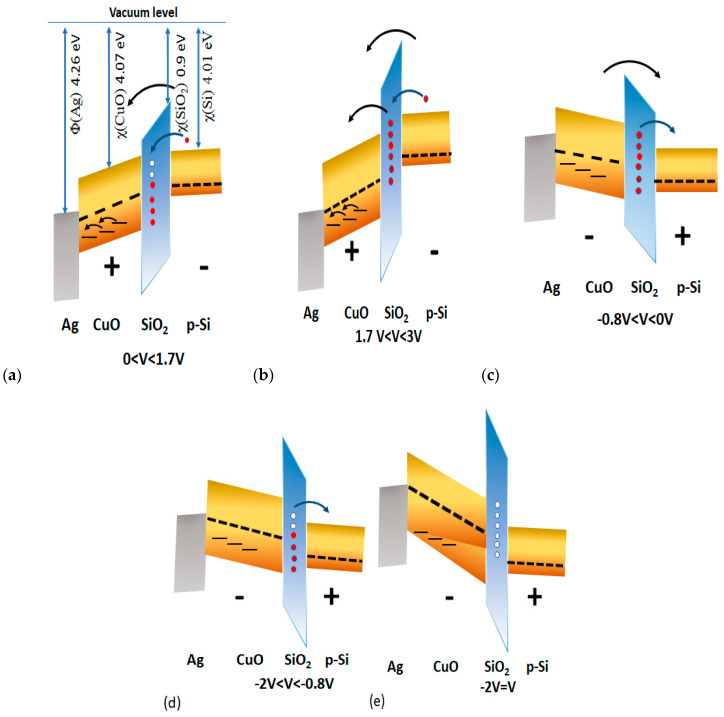
Schematic illustrating mechanism of NDR and resistive switching (RS) in Ag/CuO/SiO_2_/p-Si; (**a**–**e**) display the energy band diagram showing electric field-induced charge trapping and de-trapping in the oxide. Each figure corresponds to a sweep voltage indicated in the I-V curve.

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
