# Peer review of "Memristive Devices from CuO Nanoparticles"

_nanomaterials, 2020, doi:10.3390/nano10091677_

Round 1
Reviewer 1 Report
The authors reported the heterostructure resistive switching device composed of Ag/CuO/SiO2/p-Si layers. The authors observed the RS and NDR effect. The native oxide of Si can reduce the reset variability. The manuscript cannot be accepted because the poor English language affects the correct understanding of the work. Besides the language, the manuscript has the following drawbacks:
- The authors should carefully discuss the role of traps in CuO and the formation of Ag filament. How do these two factors contribute to the RS and NDR effect? In the manuscript, the explanation in Figure 7 is unclear.
- The I-V curves in Figure 3 and 4 are unstable. The I-V curve at 25 oC in Figure 5b has a different sweep voltage range from other curves at higher temperatures. This is not reasonable to compare them together.
- The role of SiO2 is not clear.
Author Response
Response to the Reviewer’s comments
Manuscript ID: - nanomaterials-886909
Title: Memristive devices from CuO nano-particles
We would like to thank the referee for his helpful comments. We have improved the manuscript according to his recommendations. We indeed believe that the presented results are of significant interest to the community, and hope that with our responses and amendments the referee will find our demonstrations and explanations satisfactory and consider our present version suitable for publication in nanomaterials. In the next, the reviewer’s comments to the authors are given again followed by our responses in red letters. For reviewer’s convenience, all changes are highlighted in the manuscript with yellow ink.
Comment 1.
The authors reported the heterostructure resistive switching device composed of Ag/CuO/SiO2/p-Si layers. The authors observed the RS and NDR effect. The native oxide of Si can reduce the reset variability. The manuscript cannot be accepted because the poor English language affects the correct understanding of the work. Besides the language, the manuscript has the following drawbacks.
® Response: As pointed out we have tried to improve the English to the best of our efforts.
Comment 2.
The authors should carefully discuss the role of traps in CuO and the formation of Ag filament. How do these two factors contribute to the RS and NDR effect? In the manuscript, the explanation in Figure 7 is unclear.
Response: Thank you for the comment. Comparison of the I-V characteristics of the memristive devices with and without SiO2 layers demonstrates the main role of SiO2 in the resistive switching (RS) and negative differential resistance (NDR) behavior. Therefore, the major role of traps in forming NDR is in SiO2. However, the conduction mechanism in the HRS is well described by a trap-controlled space charge limited conduction (SCLC) model. The traps in CuO forms an impurity levels which modifies the schottky barrier profile. Electric field drives the Ag ions into this levels. Further increase in field will accumulate the Ag ions which will increase the conductivity [24]. For better understanding we have updated the mechanism and Figure 7.
Page 8, line 217-233: Considering all the above results we propose a model to describe the NDR and RS behaviors which is shown in Figure 7(a)–(e). The absence of NDR in Ag/CuO/ITO provided an evidence that NDR was due to oxide trapped charges associated with defects in SiO2
At the beginning the device is HRS due to traps in CuO nanoparticles and the Schottky barrier at the Ag/CuO/SiO2/Si interface. Initially when forward voltage sweep is applied, the Schottky barrier at SiO2/p-Si interface reduces, resulting in injection of charges. This injected charges are then trapped in oxide interfacial traps [31]. Simultaneously the electric field drives the Ag ions into defect levels of CuO nanoparticles. This initiates the forward conduction, as shown in Fig. 7(a). The filled interfacial traps reduces the oxide resistance [32]. Further increase in electric field, accumulates the Ag ions which then plays a role of conductive filament. Therefore the current increases sharply with further increase in electric field owing to CF and the filled traps as shown in Figure. 7(b). The device remains in LRS until negative voltage is applied to rupture the CF. The trapped electrons will be released under bias as shown in Fig. 7(c). With a further increase of the negative voltage, the charge injection from the Ag electrode is prevented by the Schottky barrier at the interface. And it is noticed that the current decreases gradually upon increasing the sweeping voltage. This explains the NDR behaviour, as shown in Fig. 7(d). The current decreases until the trapped electrons are released completely, as shown in Fig. 7(e).
Comment 3.
The I-V curves in Figure 3 and 4 are unstable. The I-V curve at 25 oC in Figure 5b has a different sweep voltage range from other curves at higher temperatures. This is not reasonable to compare them together.
Response: Thank you for the comment. Figure 3b. shows the multiple I-V characteristics of our sample for 50 cycles. The I-V curves in Figure 3 and 4 demonstrate a good reproducibility because the difference in switching voltage is less than 10%.
Figure 5b shows the temperature dependence I-V characteristics of Ag/CuO/SiO2/p-Si. The increase in temperature shifts the current-voltage curve along the voltage axis toward lower voltages. The Vset shifts towards the lower voltage with increasing temperature. This curve provide us evidence of conduction mechanism which is space charge limited current[30]. The sweeping voltage range for all the temperature was from 3V to -2V. Due to shifting of Vset towards left of voltage axis, it resembles to be at different sweeping rate. Figure 1. shows the clear cut forward voltage sweep at all temperatures.
Figure 1. Shows the temperature dependent I-V characteristics with sweeping voltage range.
Comment 4.
The role of SiO2 is not clear.
Response: The pristine SiO2 layer contains interfacial traps[32]. It also acts as a interface barrier between CuO nanoparticle layer and p-Si. Trapping/detrapping and modulation of interface barrier results in negative differential resistance. Comparison of I-V of a device without SiO2 demonstrates the main role of SiO2 in the resistive switching (RS) and negative differential resistance (NDR) behavior. This proves that the NDR formation is due to the presence of SiO2.
Further the SiO2 layer improves the performance of device by following ways,
- The device reset involves the rupture of filaments mostly by ion migration. This sudden rupture results in scattered switching voltage distribution resulting in variation of reset voltage. Reduction in reverse current and NDR enable controlled reset, reducing the variability.
- The oxide barriers prevents the excess reverse current reducing in power consumption.
- Trapped carries in native oxide plays a prominent role in negative switching transitions. It also enables triggering of further switching at the same location which enhances controllability.

Reviewer 2 Report
Although this work is well presented and written, I have several comments about it regarding the improvement of the contribution.
- First, in my opinion the introduction should be completed with some comparison with the other memristive materials in front the CuO. And the relevance of the NDR effect in CuO in front others materials, which is the benefit.
- Again in the introduction section or the section 2, you should to explain a little bit the performance of a memristor, since the non-experienced reader could not understand your explanation!
- The spray process is done at 350ºC, can this temperature involve some impact on the quality and characteristics of the SiO2 layer? Can tihis process be done at lower temperatures?
- Line 178, you mention "...does increase the fraction of space charge in the conduction band." Could you explain it a little bit more, I so not understand is a little bit confusing for me.
- In the conclusions you mention that the device you present in this contribution reduce the device variability. Where do you see this? I have not read it anywhere!
- In figure 3 you mention that figure d is a semi-log; but I think it is a log-log figure, isn't it? Can you check it?
- To improve Fig. 2.c you can represent by using a logaritmic scale, the Cu peaks would be more clear.
- There are some typo mistakes! lines 48, 66, 86, 94, 173, 200 and 224.
Author Response
Response to the Reviewer’s comments
Manuscript ID: - nanomaterials-886909
Title: Memristive devices from CuO nano-particles
We would like to thank the referee for his helpful comments. We have improved the manuscript according to his recommendations. We indeed believe that the presented results are of significant interest to the community, and hope that with our responses and amendments the referee will find our demonstrations and explanations satisfactory and consider our present version suitable for publication in nanomaterials. In the next, the reviewer’s comments to the authors are given again followed by our responses in red letters. For reviewer’s convenience, all changes are highlighted in the manuscript with yellow ink.
Comment 1.
Although this work is well presented and written, I have several comments about it regarding the improvement of the contribution. First, in my opinion the introduction should be completed with some comparison with the other memristive materials in front the CuO. And the relevance of the NDR effect in CuO in front others materials, which is the benefits.
® Response: Thank you for putting our attention towards important point. We have updated the introduction comparing CuO with some of metal oxides such as and TiO2[1], NiO[2] and ZnO[7]. We have mention about the perovskites, and transition metal di-chalcogenide monolayers memristive devices and have highlighted some of important properties such as cost, reliability, non-toxicity and compatibility with complementary metal oxide semiconductor (CMOS) which makes CuO superior material. Memristive properties such as endurance and retention time are also being compared. The above information is provided in introduction section on page 2.
NDR along with resistive switching has additional applications which are mention in page 2, 40-43. NDR in CuO is achieved by introducing native silicon oxide and p-Si. Resistive switching along with NDR in our device has overwhelmed shortcomings in memristor such as Scattered switching, endurance deterioration during rupture of filament, high power consumption, low on-off ratio and device stability. This shortcomings are mention on page 2, line 59-60. Our device has shown improvement in the performance of memristor by overcoming all the above shortcomings.
Comment 2.
Again in the introduction section or the section 2, you should explain a little bit the performance of a memristor, since the non-experienced reader could not understand your explanation.
® Response: Thank you for bringing this to our attention. We have updated the introduction with some of basic information about memristors, in page 2. We have also provided with some of electrical parameters in result and discussion section.
Comment 3.
The spray process is done at 350ºC, can this temperature involve some impact on the quality and characteristics of the SiO2 layer? Can this process be done at lower temperatures?
® Response: Thank you for the comment. The thickness of native oxide is generally 5Å to 6.7Å [26]. As the sample was grown in oxygen rich chamber at 350Ëš C, Si gets thermally oxidize. Thermal oxidation increases the thickness of SiO2. The rate of oxide growth depends on temperature. 400ËšC is a critical point above which rate of growth is fast. Below 400ËšC the growth rate is slow. Substrate temperature has considerable effect on the morphology of sample grown in spray pyrolysis. At high temperature, the crystallinity and grain size are improved.
Comment 4.
Line 178, you mention "...does increase the fraction of space charge in the conduction band." Could you explain it a little bit more, I so not understand is a little bit confusing for me.
® Response: Thank you for the comment. Space charge limited current is the current dominated by charge carriers injected from contact. The current is dependent on mobility and not on the charge carriers[29]. The presence of traps in insulator lowers the drift mobility and also limits the free space charge carriers. When total charge is forced into the insulator as in case of trap free insulator, only the fraction of this charge is free. This value of fraction is determine by the number and depth of traps and is not dependent on applied voltage but depends on voltage in deep level traps. The final expression for space charge limit current with traps is given by
I= 10-13[V2(µÆŸ)k/d3] amperes/cm2
where µ is drift mobility and ÆŸ fraction of free carriers.
In deep level traps, space charge Q is distributed in three parts: free charges in conduction bands, trapped charge above the newly determined fermi level and trapped charge condensed in states between the original fermi level a newly determined fermi level. The change in temperature changes the steepness of traps distribution[30]. The traps distribution governs the temperature dependence. An increase in temperature does not alter the amount of space charge but does changes the traps distribution. This increases the fraction of this charge in the conduction band.
Comment 5.
In the conclusions you mention that the device you present in this contribution reduce the device variability. Where do you see this? I have not read it anywhere.
® Response: Thank you for the comment. In our manuscript we have compared Ag/CuO/SiO2/p-Si to a device without native oxide i.e Ag/CuO/ITO. The reset of device involves the rupture of filaments mostly by ion migration. This sudden rupture results in scattered switching voltage distribution resulting in variation of reset voltage [32]. The oxide barriers prevents the excess reverse current reducing in power consumption. Trapped carries in native oxide plays a prominent role in negative switching transitions. It also enables triggering of further switching at the same location which enhances controllability. Reduction in reverse current and NDR enables controlled reset reducing the device variability.
Comment 6.
In figure 3 you mention that figure d is a semi-log; but I think it is a log-log figure, isn't it? Can you check it?
® Response: Thank you for highlighting the mistake. It was typo in figure which has been updated in revised manuscript.
Comment 7.
To improve Fig. 2.c, you can represent by using a logaritmic scale, the Cu peaks would be more clear.
® Response: Thank you for the comment. In Figure 2c. we have replaced graph by using a logaritmic scale which clearly shows the Cu and O peaks.
Comment 8.
There are some typo mistakes! lines 48, 66, 86, 94, 173, 200 and 224.
® Response: Thank you for highlighting the mistake. It was typo which has been updated in revised manuscript.

Round 2
Reviewer 1 Report
Regarding Figure 5b, the response from the authors is not satisfied. The I-V curve at 25 oC provided in the response letter is different from the curve in the manuscript. The authors should check the accuracy of the data.
Author Response
Response to the Reviewer’s comments
Manuscript ID: - nanomaterials-886909
Title: Memristive devices from CuO nano-particles
We would like to thank the referee for his helpful comments. We have improved the manuscript according to his recommendations. We indeed believe that the presented results are of significant interest to the community, and hope that with our responses and amendments the referee will find our demonstrations and explanations satisfactory and consider our present version suitable for publication in nanomaterials.
Comment 1.
Regarding Figure 5b, the response from the authors is not satisfied. The I-V curve at 25 oC provided in the response letter is different from the curve in the manuscript. The authors should check the accuracy of the data.
® Response: Thank you for the comment. The reproducible I-V characteristics of our device were measured at 25 oC. In our response to review round 1, we unwittingly used one of I-V curve from our reproducible data, contrasting the manuscript figure. It can be seen that the full cycle curve in Figure (b) attached below, is similar to the Figure 5(b) in manuscript. We apologize for our mistake. So, we have updated the correct figure 5(b).
